# Myxinidin-Derived Peptide against Biofilms Caused by Cystic Fibrosis Emerging Pathogens

**DOI:** 10.3390/ijms24043092

**Published:** 2023-02-04

**Authors:** Rosa Bellavita, Angela Maione, Simone Braccia, Marica Sinoca, Stefania Galdiero, Emilia Galdiero, Annarita Falanga

**Affiliations:** 1Department of Pharmacy, School of Medicine, University of Naples ‘Federico II’, Via Domenico Montesano 49, 80131 Naples, Italy; 2Department of Biology, University of Naples ‘Federico II’, Via Cinthia, 80126 Naples, Italy; 3Department of Agricultural Sciences, University of Naples ‘Federico II’, Via dell’ Università 100, 80055 Portici, Italy

**Keywords:** cystic fibrosis, *Candida albicans*, *Achromobacter xylosoxidans*, *Stenotrophomonas maltophilia*, antimicrobial peptides, membrane interaction, polymicrobial infections, antibiofilm activity

## Abstract

Chronic lung infections in cystic fibrosis (CF) patients are triggered by multidrug-resistant bacteria such as *Pseudomonas aeruginosa*, *Achromobacter xylosoxidans*, and *Stenotrophomonas maltophilia*. The CF airways are considered ideal sites for the colonization and growth of bacteria and fungi that favor the formation of mixed biofilms that are difficult to treat. The inefficacy of traditional antibiotics reinforces the need to find novel molecules able to fight these chronic infections. Antimicrobial peptides (AMPs) represent a promising alternative for their antimicrobial, anti-inflammatory, and immunomodulatory activities. We developed a more serum-stable version of the peptide WMR (WMR-4) and investigated its ability to inhibit and eradicate *C. albicans*, *S. maltophilia*, and *A. xylosoxidans* biofilms in both in vitro and in vivo studies. Our results suggest that the peptide is able better to inhibit than to eradicate both mono and dual-species biofilms, which is further confirmed by the downregulation of some genes involved in biofilm formation or in quorum-sensing signaling. Biophysical data help to elucidate its mode of action, showing a strong interaction of WMR-4 with lipopolysaccharide (LPS) and its insertion in liposomes mimicking Gram-negative and *Candida* membranes. Our results support the promising therapeutic application of AMPs in the treatment of mono- and dual-species biofilms during chronic infections in CF patients.

## 1. Introduction

Cystic Fibrosis (CF) is a lethal autosomal recessive disease characterized by mutations in the CF transmembrane conductance regulator (CFTR) gene [1,2,3]. CFTR dysfunction produces a thick and viscous mucus layer that congests the lungs of CF patients and favors trapping of pathogens, which leads to chronic bacterial infections and inflammation [4,5]. Chronic lung infections characterized by ineffective airway functionality, colonization of multi-resistant pathogens, and progressive tissue damage are the major cause of death in CF patients [6,7]. In chronic bacterial infections, the co-existence of fungi and bacteria is very common, and despite intensive antimicrobial therapy, they are nearly impossible to eradicate due to not only the recurrence of antibiotic resistance, but also their ability to grow in microbial aggregates or biofilms [8,9,10]. Biofilms are highly organized microorganism communities formed on abiotic or biotic surfaces, where cells are entrapped in a self-produced matrix composed of extracellular polymeric substances (EPS) such as proteins, polysaccharides, and extracellular DNA.

Several bacterial and fungal species have been identified in CF patients, including *Aspergillus fumigatus*, *Candida albicans*, *Hemophilus influenzae*, *Staphylococcus aureus*, *Burkholderia cepacia* complex, *Stenotrophomonas maltophilia*, *Achromobacter xylosoxidans*, and *Pseudomonas aeruginosa* [11,12].

Although pathogens such as *P. aeruginosa*, *S. aureus*, and *H. influenzae* are the most dominant and common in CF adults, other opportunistic pathogens, such as *A. xylosoxidans* and *S. maltophilia*, are emerging. Specifically, *S. maltophilia* and *A. xylosoxidans* have been isolated from approximately 9% and 5% of CF patients, respectively [13,14]. These pathogens belong to a genus of non-fermenting Gram-negative bacteria and are characterized by a multi-drug resistance to antibiotics such as cephalosporins and aminoglycosides [15].

In particular, *Achromobacter* spp. can cause a large variety of infections, including endophthalmitis, keratoconjunctivitis, catheter-associated infections, endocarditis, pneumonia, meningitis, and peritonitis [16,17]. Among different species, *A. xylosoxidans* has been isolated in 35–80% of the sputum collected from patients with CF and is responsible for serious respiratory tract infections [18,19,20,21,22,23,24]. Similarly, significant growth and colonization of *S. maltophilia* in bronchial secretions and sputum have been observed in CF lungs [25,26,27]. Moreover, together with yeasts such as *Candida albicans*, they make the CF lungs an appropriate site for the onset of polymicrobial infections that are characterized mainly by formation of biofilms and continuous inflammation [28,29,30]. For example, several studies have shown the presence of co-infections between *P. aeruginosa* and *C. albicans* or *P. aeruginosa* and *A. fumigatus* [31,32,33], which determine the formation of strong mixed biofilms that are incredibly difficult to eradicate due to the antibiotic resistance and poor availability of effective drugs [34].

In this context, finding novel antimicrobial compounds able to inhibit and eradicate biofilms to avoid CF complications and to improve the quality of life of patients represents a sizeable challenge. Among promising alternative therapeutics, a growing interest is focused on antimicrobial peptides (AMPs) because they kill multidrug-resistant pathogens [35,36] and exhibit both anti-inflammatory and immunomodulatory activities [37,38]. Several AMPs, such as β-defensins and the cathelicidin LL-37, have been found in airway secretions of CF patients because they are produced by the respiratory epithelium [39,40], whereas other AMPs may be produced synthetically. Furthermore, both natural and synthetic AMPs have been shown to be effective against planktonic bacteria and biofilms in CF patients [41].

For example, the peptide Esculentin-1a isolated from the skin of *Rana esculenta* eradicates biofilms of *P. aeruginosa* through membrane-perturbing activity [42,43]. β-defensin analogues were developed with enhanced antimicrobial activity against *P. aeruginosa*, even with elevated salt concentrations, which is a relevant feature in CF patients [44,45]. The synthetic peptide D,L-K_6_L_9_ has shown potent antimicrobial and anti-biofilm activity against *P. aeruginosa* and high stability to degradation by sputum proteases [46]. Unfortunately, despite their strong effectiveness and potent broad-spectrum activity, the clinical applications of AMPs are limited due to their poor proteolytic stability [47,48]. Several strategies, including the introduction of unnatural amino acids [49], modifications of peptide backbone [50], and cyclizations [51], are exploited to overcome this issue.

The peptide WMR (NH_2_-WGIRRILKYGKRSK-CONH_2_) is an AMP derived from the marine Myxinidin peptide and previously developed by Cantisani et al. [52]. In previous studies, WMR showed a significant capacity both to inhibit and eradicate biofilms of *P. aeruginosa* and different *Candida* species [53,54]. Herein, to improve the proteolytic stability of WMR, we designed a series of five peptides and investigated their ability to inhibit and eradicate a five-species biofilm model comprising single- and dual-species biofilms of three emergent CF microorganisms (*C. albicans*, *S. maltophilia*, *A. xylosoxidans*). In addition, the activity of WMR-4, the most active analogue, on survival rate and virulence was evaluated in vivo using *Galleria mellonella* larvae. The hypothetical mechanism of action of the peptide WMR-4 was examined through biophysical studies such as fluorescence measurements and circular dichroism (CD) spectroscopy.

## 2. Results

### 2.1. WMR Serum Stability and Design of Novel Analogues

Before investigating the activity of WMR analogues against pathogens implicated in CF chronic infections, we reasoned upon the improvement of the proteolytic stability of the native peptide WMR by identifying the most abundant peptide fragments after incubation with 50% fresh bovine serum within 4 h. The percentage of intact WMR was monitored by calculating the peak area of the chromatogram from the RP-HPLC analysis; furthermore, the deleted peptide fragments were analyzed through electrospray ionization mass spectrometry (ESI-MS). As reported in Figure 1, 68% of WMR was intact after 1 h, and only 20% was intact after 5 h. The peptide fragments identified with a mass of 1443.86 Da, 1543.86 Da, and 1361.80 Da (Table 1) corresponded to the native sequence WMR without (i) Lys^14^, (ii) Lys^14^ and Ser^13^, and (iii) Lys^14^, Ser^13^ and Trp^1^, respectively.

On the basis of the proteolytic degradation of WMR, we thus designed five different peptides (Table 2).

Lys^14^ and Ser^13^ were replaced with their D-enantiomers, yielding peptides WMR-1 and WMR-2, respectively. Gly^2^ was substituted with the unnatural amino acid 2-aminoisobutyric acid (Aib), yielding peptide WMR-3. Aib was selected because it reduces susceptibility to proteolytic cut and favors the helical structure [55,56], which is key for the biological activity of WMR, as previously shown by Cantisani et al. [52]. Furthermore, D-enantiomers and Aib modifications were combined, yielding peptide WMR-4. In addition, Lys^14^ was replaced by the unnatural basic amino acid L-2,3-diaminopropionic acid (Dap), yielding WMR-5.

### 2.2. Minimal Inhibitory Concentration (MIC)

The susceptibility of the three different strains (*C. albicans* ATCC 90028, *A. xylosoxidans* DSM 2402, and *S. maltophilia* DSM 50170) which are implicated in CF chronic infections to WMR and its analogues, fluconazole (FLC), and meropenem (MEM) is shown in Table 3. All the WMR analogues, except the peptide WMR-4, showed an MIC > 50 μM as the native peptide towards *C. albicans* ATCC 90028. However, the activity of the analogue WMR-4 was slightly improved, reaching an MIC value of 50 μM.

The MIC values obtained against the two Gram-negative bacteria (*A. xylosoxidans* DSM 2402 and *S. maltophilia* DSM 50170) decreased from 50 μM to 25 μM for the peptides WMR-4 and WMR-5, showing a more significant antibacterial activity compared to the peptide WMR. Peptides WMR-1, WMR-2, and WMR-3 showed an MIC > 50 μM against both Gram-negative strains. In addition, the susceptibility of *C. albicans* to FLC, as well as that of *A. xylosoxidans* and *S. maltophilia* to MEM, is reported in Table 3.

This initial antimicrobial screening on *C. albicans*, *A. xylosoxidans*, and *S. maltophilia* prompted us to select the peptide WMR-4 as the best candidate for our studies.

### 2.3. Serum Stability of the Analogue WMR-4

The proteolytic stability of WMR-4 was evaluated by incubation of the peptide with 50% fresh bovine serum within 16 h, and the degradation process was monitored by RP-HPLC. As expected, the peptide WMR-4 displayed significantly superior stability compared to WMR (Figure 2). Specifically, we observed that 70% of the intact WMR-4 was present after 5 h of incubation, compared with 20% of the native peptide WMR at the same time point. Moreover, about 30% of WMR-4 was detected after an incubation of 16 h. This result indicated that the introduction of residues Aib^2^, DSer^13^, and DLys^14^ protected the peptide WMR-4 from degradation by serum proteases.

### 2.4. Biofilm Formation Assay and Antibiofilm Activity

The extent of biofilm formation (Figure 3a,b) among the two bacteria was very high, allowing them to be considered as strong biofilm producers. Otherwise, the biofilm producing ability of *C. albicans* was classified as moderate. To characterize the vital biomass of mono- and dual-species biofilms, viable cells of the three microorganisms were counted by colony forming units per well (CFU well-1), and results are reported in Figure 3,b. However, in the first 24 h, the vital biofilm biomass of *C. albicans*, *S. maltophilia*, *A. xy-losoxidans*, and the two mixed biofilms (*CAx*, *CSm*) was 7.0, 6.1, 7.1, 7.7, and 7.8 log CFU/well, respectively; at 48 h, there was an increase in about 1 log CFU/well for all biofilms. The mixed biofilm *CAx* was composed of 45% *C. albicans* and 55% *A. xylosoxidans* at 24 h, and it was composed of 36% *C. albicans* and 64% *A. xylosoxidans* at 48 h. The mixed biofilm *CSm* was composed of 60% *C. albicans* and 40% *S. maltophilia* at 24 h, and it was composed of 27% *C. albicans* and 73% *S. maltophilia* at 48 h, respectively.

To determine the inhibitory effect of WMR-4 on biofilm formation (Figure 4), the biofilm-forming abilities of *C. albicans*, *S. maltophilia*, *A. xylosoxidans*, and the two mixed biofilms *CAx* (*C. albicans* and *A. xylosoxidans*) and *CSm* (*C. albicans* and *S. maltophilia*) were determined in the presence of variable concentrations of WMR-4. After incubation with the peptide, the biofilm was analyzed via staining with crystal violet.

The inhibition of biofilm formation was found to be concentration dependent. As shown in the Figure 4, sub-MIC concentrations of WMR-4 in the range of 2.5 to 20 µM inhibited biofilm formation. Biofilm formation was similarly affected in the case of both bacterial strains, and it reached about 100% inhibition at the highest concentration tested, which was 20 µM. WMR-4 increased the inhibition of the *C. albicans* biofilm by up to 80% at a concentration of 20 μM, as well as that of the two mixed biofilms, in which 75% inhibition was achieved at the highest concentration tested.

As shown in Figure 5, WMR-4 was also involved in the disruption of preformed mono- and dual-species bio-films. After treatment of the established biofilm with the sub-MIC of 20 μM, the *C. albicans* biofilms decreased by 80%, whereas the two bacteria biofilms decreased by 60%. The peptide WMR-4 showed a moderate capacity to eradicate mature mixed biofilms, causing an eradication of 40% for *Cax* and 50% for *Csm*.

### 2.5. RT-PCR

Molecular responses of each microorganism in the mono- and dual- species biofilms to WMR-4 were investigated by qPCR. Because WMR-4 had a significant inhibition activity on biofilm formation for *C. albicans*, *S. maltophilia*, and *A. xylosoxidans* and for the mixed biofilms (*CAx*, *CSm)*, we investigated the hypothetical involvement of WMR-4 in regulating the expression of some genes associated with stress, biofilm, and quorum-sensing signaling. For the transcriptional alterations of *C. albicans*, expression of hyphal adhesin *ALS3* and ergosterol biosynthesis enzyme *ERG11*, which are involved in the adherence [57], and *HOG1*, which plays a key role in virulence and response to stress, were selected [58]. As shown in Figure 6, *ALS3* was significantly down-regulated in all three conditions, whereas *ERG11* and *HOG1* were slightly upregulated in both mixed biofilms. These data indicated that WMR-4 counteracted *C. albicans* in mono- and dual-species biofilms by exerting effects on *Candida* adhesins. The upregulation of the β-1,3-D-glucan synthase showed that WMR-4 did not interfere with the synthesis of the fungal cell wall, whereas the upregulation of *HOG1* involved in the oxidative stress response showed its activation in response to this antifungal in both mixed conditions.

We selected for *S. maltophilia:* (i) the gene *smf-1*, which encodes for type-1 fimbriae, which plays a significant role in adherence to surfaces and in the early stages of biofilm formation; (ii) the gene *rpfF*, which plays a critical role in the production of the diffusible signal factor (DSF), which mediates quorum sensing in *S. maltophilia* [59]; and (iii) *Ax21*, which works as a quorum-sensing factor that significantly impairs motility, induces inhibited biofilm formation, and reduces virulence [60]. The remarkable downregulation of genes *smf1*, *rpfF*, and *Ax21* during inhibition of single- and dual-biofilm formation of *S. maltophilia* showed the involvement of these genes in both conditions, confirming that the treatment by WMR-4 critically influenced several biosynthetic pathways and signaling systems. As for *A. xylosoxidans*, several genes were detected: (i) the *epsF* gene, which encodes exopolysaccharide biosynthesis; (ii) *AxyA* multidrug resistance efflux pump genes; and (iii) *UspA*, a universal stress protein [61]. The expression of *epsF* gene was undetectable in mono and dual-species biofilms; on the contrary, the expressions of *AxyA* were significantly downregulated in both conditions. Notably, as generally expected, the stress protein *UspA* was upregulated in both conditions.

### 2.6. In Vivo Toxicity and Activity of the Peptide on the Infected G. mellonella

The in vivo effect of WMR-4 at different peptide concentrations of 5, 10, and 20 μM was evaluated using a *G. mellonella* model system [62]. As shown in Figure 7, larvae showed about 80% survival at up to 96 h of observation with respect to intact larvae, indicating that WMR-4 was not toxic at the three concentrations tested. As shown in Figure 8, the infection induced in *G. mellonella* with *C. albicans*, *A. xylosoxidans*, and *S. maltophilia* at concentrations of 10^6^ cells/larvae decreased the number of living larvae by 80%, 90%, and 80%, respectively, after 96 h. However, for the mixed infection with dual species (*CAx*, *CSm*), there was no survival for *CAx* and only 20% for *CSm* at 96 h. When we treated larvae with 10 μM of WMR-4 before or after infection, we observed an increase in survival by about 20%, and the survival reached about 80% at 96 h. It was more evident in larvae pretreated with WMR-4 and each microorganism alone than when microorganisms were used in pairs.

### 2.7. Peptide Secondary Structure

To explore the influence of amino acid replacement on peptide secondary structure, we performed circular dichroism (CD) spectroscopy for peptides WMR and WMR-4 under different conditions, including water and 20%, 40%, and 60% of 2,2,2–trifluoroethanol (TFE). As evidenced in Figure 9, both peptides adopted a random-coil conformation in water, whereas a different behavior was observed in the presence of TFE. The peptide WMR-4 exhibited an α-helical conformation already at the lowest percentage (20%) of TFE, and this helical content increased in the presence of both 40% and 60% of TFE up to 15%. However, the peptide WMR adopted a disordered conformation in 20% TFE and a helix conformation only in the presence of higher percentages of TFE (40% and 60%). These results indicated that the replacement of Gly^2^ with the unnatural amino acid Aib induced and increased the helical content of the peptide WMR-4 in comparison with the native peptide WMR.

### 2.8. Interaction of WMR-4 with Lipopolysaccharide (LPS)

The mode of action of the peptide WMR-4 was investigated in the presence of lipopolysaccharide (LPS), which is exposed on the outer membrane of Gram-negative bacteria and is generally the main agglutination site of AMPs [63]. Firstly, we measured the ability of WMR-4 to aggregate in the presence of LPS (100 μg/mL) at different peptide concentrations of 20 μM, 30 μM, and 50 μM and when Thioflavin T (ThT) was used as a fluorescent probe (Figure 10A).

As detected in Figure 10A, we did not observe a significant increase in ThT emission, indicating that WMR-4 did not oligomerize in the presence of LPS. Furthermore, we performed a tryptophan (Trp) quenching experiment to verify the interaction with LPS (Figure 10B,C). The Trp residue is used as a probe to investigate the localization of the peptide within the membrane, as its fluorescence emission increases when it is located in a more hydrophobic environment. In our study, we evaluated the interaction of WMR-4 with LPS through quenching of the accessible Trp^1^ in the presence of acrylamide [64] at different concentrations and recorded its change in the fluorescence emission.

Furthermore, we evaluated the Trp emission of the peptide WMR-4 in water (Figure 11). As shown, WMR-4 interacts strongly with acrylamide in water. The calculated Stern–Volmer (Ksv) quenching constant is 15.4 ± 1.3 in water and 6.48 ± 0.90 in LPS, clearly confirming that the Trp is more buried in LPS.

### 2.9. WMR-4: Liposome Insertion and Leakage

We also investigated the mode of action of WMR-4 in the presence of large unilamellar vesicles (LUVs), which mimic Gram-negative (DOPE/DOPG/CL) and *C. albicans* (PE/PC/PI/Erg) membranes. Similarly to what was observed in LPS, the peptide WMR-4 did not aggregate in LUVs mimicking Gram-negative and *C. albicans* membranes, but WMR-4 was buried in both LUVs, as well as in LPS. Specifically, we performed the Trp quenching experiment, as described above, to monitor the insertion of WMR-4 in liposomes. In DOPE/DOPG/CL, the Trp^1^ was completely quenched, and the low Ksv value of 2.20 ± 0.29 confirmed the insertion of WMR-4 in LUVs mimicking which mimic Gram-negative membrane (Figure 12A,B). In addition, we analyzed the capacity of WMR-4 to induce liposome leakage after its insertion in lipidic bilayers by using 8-aminonaphtalene-1,3,6-trisulfonic acid, disodium salt (ANTS), and p-xylene-bis-pyridinium bromide (DPX) as fluorescent probes. As shown in Figure 12C, a low leakage percentage (~20%) was observed even at the highest concentration of 50 μM. Thus, we hypothesized that the peptide WMR-4 had a good capacity to interact and insert in LUVs mimicking that mimic Gram-negative membranes and induced a low percentage of leakage, as observed previously for the native peptide WMR [52].

Moreover, regarding the mode of action in the presence of LUVs mimicking *Candida* membrane, WMR-4 had the same behavior observed in LUVs of DOPE/DOPG/CL. In fact, Trp^1^ was partially accessible to acrylamide quenching, and the peptide WMR-4 was buried in LUVs of PE/PC/PI/Erg, as shown in Figure 13. In addition, we performed the leakage assay in the same condition described above, and we did not observe LUV leakage in the concentration range from 5 μM to 50 μM. Thus, we hypothesized that the peptide WMR-4 interacts with the fungal membrane, causing disruption through a carpet-like or other molecular mechanisms that will be further investigated.

## 3. Materials and Methods

### 3.1. Peptide Synthesis

Peptides were synthesized using a standard solid-phase Fmoc (9-fluorenylmethoxycarbonyl) method, as previously reported [65]. Briefly, a rink amide resin was used, the removal of the Fmoc protecting group on the resin and amino acids was performed with 20% (*v/v*) piperidine in DMF (0.5 + 1 min, under ultrasonic irradiations), and the couplings were carried out in the presence of 3 equiv. Fmoc-protected amino acid, HOBt (3 equiv.) and HBTU (3 equiv.), and 6 equiv. DIPEA as the basis (2 × 10 min, under ultrasonic irradiation) [66]. All crude peptides were obtained with good yields (70–80%). At the end of the synthesis, peptides were cleaved from the resin with an acid solution of trifluoroacetic acid (TFA), precipitated in ice-cold diethyl ether, purified by preparative reversed-phase high-pressure liquid chromatography (RP-HPLC) equipped with a Phenomenex Jupi-ter 4 μm Proteo 90 Å 250 × 21.20 mm column with a linear gradient of solvent B (0.1% TFA in acetonitrile) in solvent A (0.1% TFA in water) from 5 to 70% in 20 min with UV detection at 210 nm, and analyzed through ESI-MS (see Appendix A).

### 3.2. Strains and Cultural Conditions

*Candida albicans ATCC 90028*, *Achromobacter xylosoxidans DSM 2402*, and *Stenotrophomonas maltophilia DSM 50170* were used in this study. Microbial strains were grown in Tryptone Soya Broth (TSB, VWR chemicals, Leuven, Belgium), supplemented with 0.1% glucose only for *C. albicans* at 37 °C overnight. Then, cells were harvested by centrifugation at 5000 rpm, 4 °C for 10 min, followed by washing in Phosphate Buffered Saline (PBS, Oxoid Ltd., Basingstoke, UK) three times. Standardized 1 × 10^6^ CFU mL**^−^**^1^ inoculum was used in the next experiments.

### 3.3. Minimum Inhibitory Concentration (MIC)

The minimum inhibitory concentrations (MIC) of WMR-4 against *C. albicans*, *A. xylosoxidans,* and *S. maltophilia* were determined using a broth microdilution method accord-ing to CLSI M27-A3 and M07-A9 guidelines [67,68]. Concentrations of WMR-4 from 0.5 to 50 µM were added to wells of a 96-well microplate containing 1 × 10^6^ CFU mL^−1^ cells. Fluconazole at a concentration range from 1 to 50 µM and meropenem at a concentration range from 10 to 200 µM were used as control. The plate was incubated at 37 °C for 24/48 h due to poor growth after 24 h for the two bacterial species. Next, the absorbance at 590 nm was read using a microplate reader (Synergy H4; BioTek Instruments, Agilent Technologies, Winooski, VT, USA). MIC value was defined as a lowest concentration of WMR and analogues able to inhibit total microbial growth.

### 3.4. Proteolitic Stability

The proteolytic stability of WMR and WMR-4 were determined in bovine serum acquired from ThermoFisher Scientific (Milan, Italy). Peptides were dissolved in sterile water to prepare stock solutions, then incubated with 50% of bovine serum to obtain a final concentration of 200 μM [69]. The mixture peptide/serum was incubated at 37 ± 1 °C for different times (0.25, 0.50, 1, 2, 3, 4, 5, and 16 h). At each pre-established time, an aliquot of reaction was taken, and acetonitrile was added to favor the precipitation of serum proteins. Then, the solution was cooled at 4 °C and centrifuged for 15 min at 13,000× *g* rpm. The supernatant was analyzed by HPLC in a Phenomenex Jupiter 4 μm Proteo 90 Å 250 × 21.20 mm column with a linear gradient of solvent B (0.1% TFA in acetonitrile) in solvent A (0.1% TFA in water) from 5 to 70% in 20 min with UV detection at 210 nm and a flow rate of 1 mL/min.

### 3.5. Biofilm Formation Assay

The ability to form mono- and dual-species biofilm of *C. albicans*, *A. xylosoxidans* and *S. maltophilia* was evaluated using a 96-well microplate. *A. xylosoxidans* and *S. maltophilia* were grown in TSB at 37 °C for 24 h, the pellet was washed in phosphate-buffered saline (PBS), and the cells were resuspended in RPMI 1640 medium (ThermoFisher Scientific) supplemented with 3-(N-morpholino) propanesulfonic acid (MOPS) (pH 7.3). Afterward, 100 μL cell suspension was seeded into each well and incubated at 37 °C for 24/48 h. *C. albicans* was grown in TSB 1% glucose at 37 °C for 24 h. Cell suspensions were adjusted to the value of 1´10^6^ cells in RPMI 1640 medium supplemented with MOPS (pH 7.3), and 100 μL was seeded into each well of the microtiter plates and incubated at 37 °C for 24/48 h. After pellet collection, dual-species biofilms of *C. albicans/A. xylosoxidans* (Mix*CAx*) and *C. albicans/S. maltophilia* (Mix*CSm*) were obtained when the two microorganisms in RPMI 1640 medium supplemented with MOPS were mixed at a ratio of 1:1 and 100 µL of the suspension was in each well. Culture media without cells represent the negative control. Plates were incubated for 48 h at 37 °C. Total biofilm biomass was detected by crystal violet (CV) assay, as per the guidelines of Stepanović et al. with some modifications [70,71]. Briefly, the plates were washed three times with PBS, and the biofilm was fixed at 37 °C for 1 h. In total, 200 µL of CV (0.2% *v*/*v*) was added to well, and after 15 min, the excess was washed with PBS, and the biofilm was resuspended by adding 300 µL of acetic acid at 32% (*v/v*). The absorbance at 570 nm was read using a microplate reader (Synergy H4; Bio-Tek Instruments, Agilent Technologies, Winooski VT05404 USA). Each experiment was repeated in triplicate.

ODcut-off (ODc) was calculated to classify the ability of microorganisms to form bio-film using the following formula: OD570 mean of negative control plus three times the standard deviation (SD). The classification was as follows: negative (OD ≤ ODc), weak (ODc ≤ OD ≤ 2 × ODc), moderate (2 × ODc < OD), or strong (OD ≥ 4 × ODc).

To determine the vital biofilm biomass, an assay of colony-forming units (CFUs) was used.

The biofilm was washed three times to remove non-adherent cells, scraped from the bottom of the well, vigorously vortexed, and resuspended in PBS. An amount of 100 µL of biofilm suspension was plated on Tryptone Soya Agar (TSA, VWR chemicals, Leuven, Belgium), which was supplemented with amphotericin B for bacterial strains, and Rosa Bengal agar (RB, Sigma-Aldrich, St. Louis, MO, USA), which was supplemented with chloramphenicol for C. albicans. The CFUs were counted after 24–48 h of incubation at 37 °C.

### 3.6. Antibiofilm Activity

The effect of WMR-4 to inhibit single and mixed-species biofilms was evaluated by adding scalar concentrations of peptide, ranging from 5 to 20 µM, to microbial inoculum. The plates were incubated at 37 °C for 48 h. After incubation time, biofilm biomass was evaluated using CV, as reported above.

The influence of WMR-4 on preformed biofilms was assessed as follows: 48 h biofilms were treated with serial concentrations of the peptide (5 to 20 µM), and the plates were incubated for another 24 h at 37 °C. The ability to eradicate biofilm was evaluated by CV, as previously described.

Biofilm reduction was calculated as [(OD_570_ of the control − OD_570_ of the treated)/OD_570_ of the control] × 100.

### 3.7. Quantitative Real Time PCR

The potential mechanisms of WMR-K4 to inhibit biofilms of *C. albicans*, *A. xylosoxi-dans*, and *S. maltofilia* was investigated trough gene expression analysis of genes involved in virulence or stress response using qRT-PCR. RNA extraction and real-time PCRs were performed as described previously [72]. For *C. albicans*, ERG11, ALS3, and HOG1 were selected; for *A. xylosoxidans epsF*, *AxyA*, and *UspA* and for *S. maltofilia rpfF*, *smf-1*, and *Ax21* genes [73]. Actin and 16S RNA were used as control. All primers used in these studies are shown in Table 4.

Cells of single or mixed biofilm grown together with WMR-4 (10 µM) at 37 °C for 24 h were scraped and washed in PBS, and the pellets were used to extract total RNA according to the manufacturer’s protocol of DirectzolTM RNA Miniprep Plus Kit (ZY-MO RESEARCH, Irvine, CA, USA). Nanodrop spectrophotometer 2000 (Thermo Scientific Inc., Waltham, MA USA) was used to determine the concentration and the purity of extracted RNA.

An amount of 1000 ng of RNA was retrotranscribed with the QuantiTect Reverse Transcription Kit (Qiagen, Valencia, CA, USA). The QuantiTect SYBR Green PCR Kit (Qiagen, Valencia, CA, USA) was used to perform Real-Time PCR. Briefly, 25 µL of SYBR Green was added to 100 ng of cDNA, 1 µM of each primer, 12.5 µL of QuantiFast SYBR Green PCR Master Mix (2×). PCR cycling was performed as described in a previous study [54]. RT-PCR was performed in a Rotor-Gene Q cycler (Qiagen, Valencia, CA, USA). The Pfaffl method was used to analyze and normalize the values of expression of each gene [74,75].

### 3.8. In Vivo Effect of WMR-4 on Galleria mellonella Larvae

The *G. mellonella* microorganism infection model was used to evaluate the antimicrobial effect of WMR-4 in vivo, and a survival assay was performed as described previously [76,77]. Briefly, to test toxicity of WMR-4, larvae were treated with different concentrations (5, 10, 15, and 20 µM) of peptide, and survival was monitored every 24 h for a duration of 4 days.

The non-toxic concentration of peptide was selected to conduct experiments of prevention and therapeutic treatment of infection by *C. albicans*, *A. xylosoxidans*, or *S. maltofilia* alone or in combination (Mix*CAx* and Mix*CSm*). Larvae were infected using an inoculum of 1 × 10^6^ CFU mL^−1^. Infected larvae were treated with WMR-4 at a concentration of 10 µM, 2 h pre-infection (prophylactic treatment) or 2 h post-infection (therapeutic treatment), and were incubated at 37 °C for 96 h. Every 24 h, the survival was monitored. Each experiment used groups containing 20 larvae. The groups included groups of non-injected larvae and groups of larvae injected with PBS, which served as control, and experiments were repeated twice.

### 3.9. Statistical Analyses

Statistical analyses were executed using GraphPad Prism Software (version 8.02 for Windows, GraphPad Software, La Jolla, CA, USA, www.graphpad.com, accessed on 3 December 2022). Data are the result of three independent experiments, which are represented as mean ± standard deviation (SD). The Kaplan–Meier method was used to plot survival curves. For a multiple comparison test, one-way ANOVA following Tukey’s test was used. Asterisks show significant differences, (* = *p* < 0.05, ** = *p* < 0.01, *** = *p* < 0.001, **** = *p* < 0.0001).

### 3.10. Circular Dichroism (CD) Spectroscopy

CD spectra of peptides WMR and WMR-4 at a concentration of 70 μM were recorded using quartz cells of 0.1 cm path length at 25 °C. CD spectra were performed in aqueous solvent and in the presence of 20%, 40%, and 60% TFE. Each spectrum was measured in the 260–196 nm and converted into mean molar ellipticity.

### 3.11. Liposome Preparations

Large unilamellar vesicles (LUVs) mimicking Gram-negative bacteria (DOPE/DOPG/CL, 63/25/12, ratio in moles) and *Candida* (PE/PC/PI/Ergosterol, 5:4:1:2, ratio in moles) were prepared according to the extrusion method, as previously reported [78,79,80]. LUVs were loaded with ANTs and DPX fluorescent probes to monitor their leakage induced by WMR-4. Briefly, the dry lipid films were dissolved with a solution of ANTs (12.5 mM) and DPX (45 mM) in water and then lyophilized overnight. The lipid films with encapsulated ANTS and DPX were hydrated with PBS 1× buffer; the lipid suspension was freeze-thawed 6 times and extruded 10 times through polycarbonate membranes with 0.1 µm diameter pores to obtain LUVs.

Regarding peptide aggregation in LUVs, the lipid films were hydrated with 100 mM NaCl, 10 mM Tris-HCl, 25 µM Tht buffer, pH 7.4, vortexed for 1 h, and treated to obtain LUVs [81].

### 3.12. Peptide Aggregation by Thioflavin T Assay

The aggregation of peptide WMR-4 in the presence of LPS and liposomes that mimic Gram-negative and *Candida* membranes was measured using Thioflavin T as a fluorescent probe. WMR-4 was incubated with LPS (100 μg/mL) and ThT (25 μM) at the concentrations of 20, 30, and 50 μM for 1 h.

Firstly, lipid films were prepared at the concentrations of 100 μM and 50 μM for Gram-negative and *Candida* liposomes, respectively. Then, LUVs were titrated with different peptide concentrations of 5, 10, 15, and 20 μM. Each ThT emission spectrum was recorded at a fluorescence excitation of 450 nm (slit width, 10 nm) and a fluorescence emission of 482 nm (slit width, 5 nm) before and after the peptide addition [82,83].

### 3.13. Liposomes Leakage by ANT/DPX Assay

The liposome leakage was evaluated using ANTS and DPX as fluorophores [84,85]. Lipid films that mimic Gram-negative (final concentration = 100 μM) and *Candida* (final concentration = 50 μM) membranes with encapsulated fluorophores (12.5 mM ANTS and 45 mM DPX) were prepared as described above [78]. LUVs were titrated with peptide concentrations of 5, 10, 15, 20, and 50 μM. Each ANTS emission spectrum was measured by setting an excitation fluorescent at 385 nm (slit width, 5 nm) and a fluorescent emission at 512 nm (slit width, 5 nm). The percentage of leakage was calculated as: %leakage = (F_i_–F_0_)/(F_t_ – F_0_), where F_0_ is the intensity of ANTS fluorescence before the peptide treatment, F_t_ is the fluorescence after the treatment with Triton-X (10% *v*/*v*) to obtain a complete liposome leakage, and F_i_ is the intensity after the addition of the peptide.

### 3.14. Tryptophan Quenching by Acrylamide

The tryptophan (Trp) insertion in LPS and in LUVs mimicking Gram-negative and *Candida* membranes was monitored using acrylamide as the Trp quencher [86]. Specifically, the tryptophan residue of WMR-4 (peptide concentration = 5 μM), both in water and incubated with LPS (100 μg/mL) and LUVs (C_f_ = 200 μM), was quenched by acrylamide at different concentrations of 0.02, 0.04, 0.06, 0.08, 0.1, 0.12, and 0.16 M [79]. The Trp emission spectrum was measured at a fluorescence excitation of 295 nm and a fluorescence emission of 340 nm to eliminate interference from the Raman band of water [87]. The data were analyzed using the Stern–Volmer equation F_0_/F = 1 + K_sv_ [Q], where F_0_ is the fluorescence intensity in the absence of the quencher (Q), F is the fluorescence intensity in the presence of quencher, and K_sv_ is the Stern–Volmer quenching constant. This constant is a reliable reflection of the bimolecular rate constant for collisional quenching of the tryptophan residue present in the aqueous phase.

## 4. Conclusions

The inadequacy and deficiency of antibiotics for treating CF patients have led researchers to consider alternative strategies to fight lung infections characterized by polymicrobial infections. Biofilms are considered to be an important factor in the microorganism’s ability to cause diseases. The composition and diversity of these infections change throughout the patient’s lifetime. Herein, we designed and developed a more serum-stable version of the peptide WMR, namely WMR-4, which is able to inhibit and eradicate single- and dual-species biofilms of three emergent CF microorganisms, such as *C. albicans*, *S. maltophilia*, and *A. xylosoxidans*. The serum-stability results showed that peptide WMR-4, which bears D-enantiomers (D-Ser^13^ and D-Lys^14^) and Aib modifications, emerged as the most stable compound to the proteolytic cut because about 70% of the intact peptide was detected after 5 h of incubation. The biological data showed that the MIC values of the peptide WMR-4 against the two Gram-negative bacteria (*A. xylosoxidans* DSM 2402 and *S. maltophilia* DSM 50170) decreased from 50 μM to 25 μM, showing a better antibacterial activity than that of the native peptide WMR, whereas an MIC of 50 μM was observed on *C. albicans* ATCC 90028. In addition, WMR-4 increased the inhibition of *C. albicans* biofilm by up to 80% at a concentration of 20 μM, as well as for the two mixed biofilms *Cax* and *Csm*, in which 75% inhibition was achieved at the same concentration. Concerning the eradication activity, WMR-4 displayed a moderate capacity to eradicate mature mixed biofilms, causing an eradication of the two dual-species biofilms of 40% *Cax* and 50% *Csm*, respectively. Moreover, the peptide WMR-4 was found to be non-toxic in an in vivo model using *G. mellonella* and increased the survival of infected larvae. We also investigated the mode of action of the peptide WMR-4 through biophysical studies, showing that WMR-4 is unable to aggregate in LPS, and in liposomes mimicking Gram-negative and fungi membranes, whereas its Trp residue is more inserted in Gram-negative and fungi membranes than in LPS. The absence of leakage led us to hypothesize not only that the mechanism of action is likely to be a carpet model, but also that other intracellular mechanisms may be involved. In conclusion, the present work reveals the potential therapeutic applications of the antimicrobial peptides for fighting polymicrobial infections present in CF patients.

## Figures and Tables

**Figure 1 ijms-24-03092-f001:**
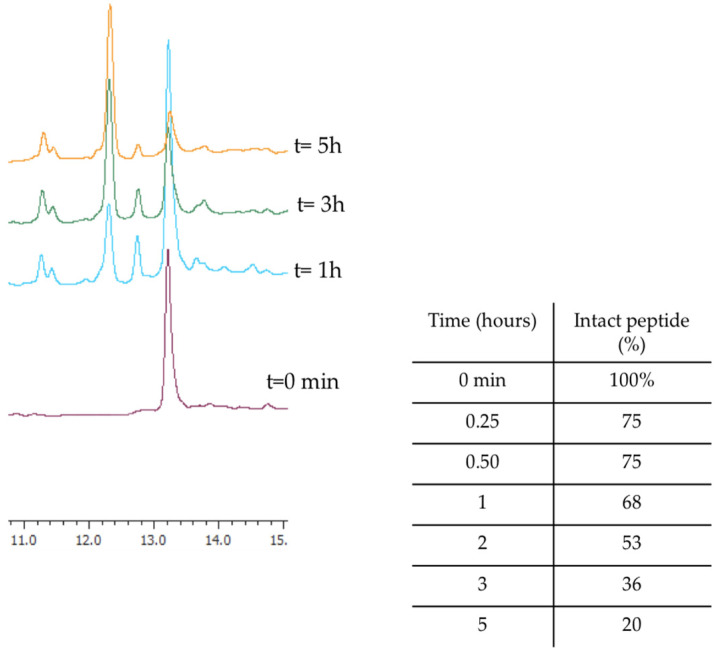
The representative RP-HPLC chromatograms of WMR at different times (0, 1, 3, and 5 h), and the percentage of intact peptide after incubation with 50% bovine serum at 37 °C.

**Figure 2 ijms-24-03092-f002:**
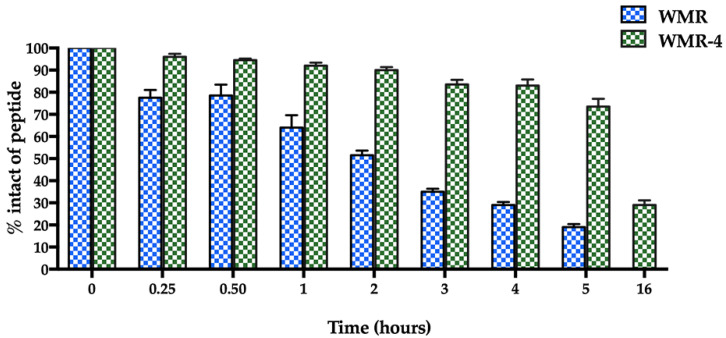
The serum stability of peptides WMR and WMR-4 in bovine serum determined by RP-HPLC. The percentage of intact peptide was obtained by calculating the relative peak area monitored by RP-HPLC and subtracting background peak areas in blank matrix.

**Figure 3 ijms-24-03092-f003:**
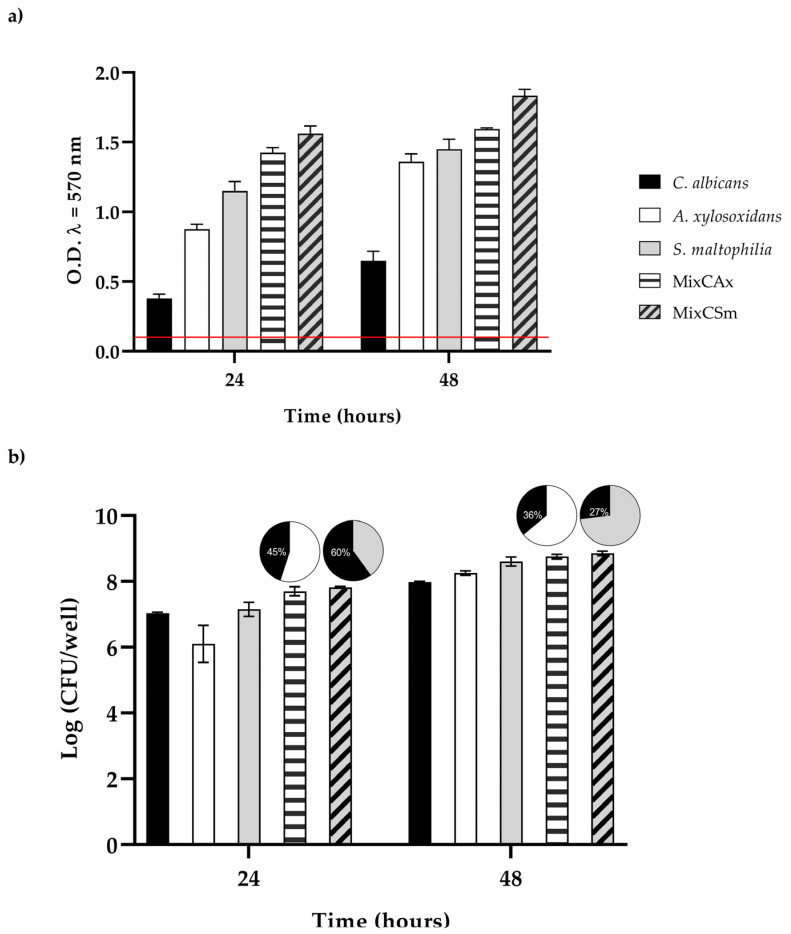
(**a**) Biofilm formation capacity of *C. albicans*, *A. xylosoxidans*, *S. maltophilia*, and the two mixed biofilms (*CAx*, *CSm*) using the crystal violet staining method. Red line represents ODcut. ODcut = mean of negative control with addition of 3 times the SD. (**b**) Enumeration of colony forming units per well for single- and dual-species biofilms.

**Figure 4 ijms-24-03092-f004:**
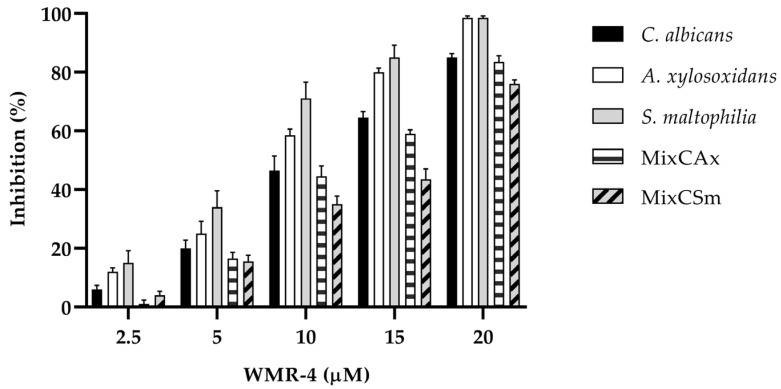
Antibiofilm activity of WMR-4 on *C. albicans*, *S. maltophilia*, *A. xylosoxidans*, and two mixed biofilms (*CAx*, *CSm*) quantified with crystal violet after 24 h.

**Figure 5 ijms-24-03092-f005:**
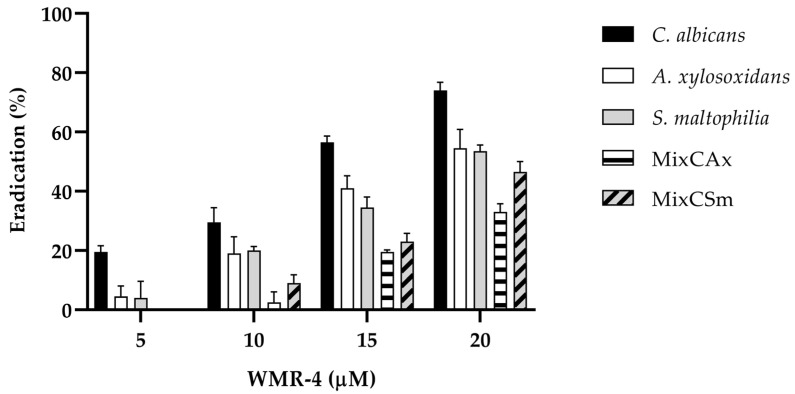
Disruption of established biofilms of *C. albicans*, *S. maltophilia*, and *A. xylosoxidans* and two mixed biofilms (*CAx*, *CSm*) after the treatment with the peptide WMR-4.

**Figure 6 ijms-24-03092-f006:**
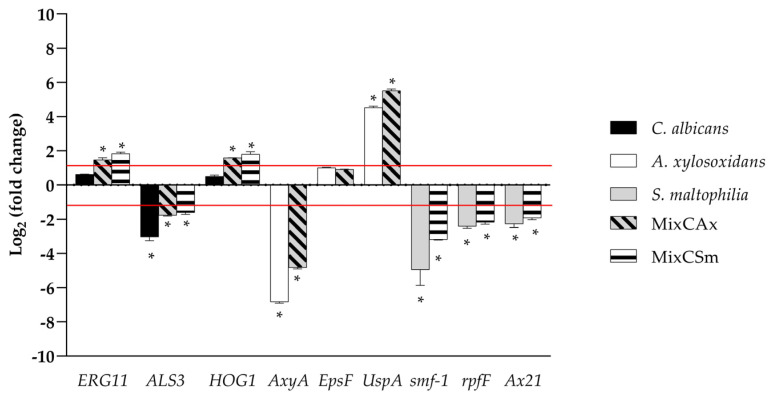
Real-time qPCR during inhibition of single and mixed biofilm using WMR-4 at concentration of 10 µM. Histograms represent the fold differences in the expression levels of the genes selected during inhibition of single and mixed biofilm with WMR-4 at concentration of 10 µM. Red lines show fold change thresholds of −1 and +1, respectively. * = *p* < 0.05.

**Figure 7 ijms-24-03092-f007:**
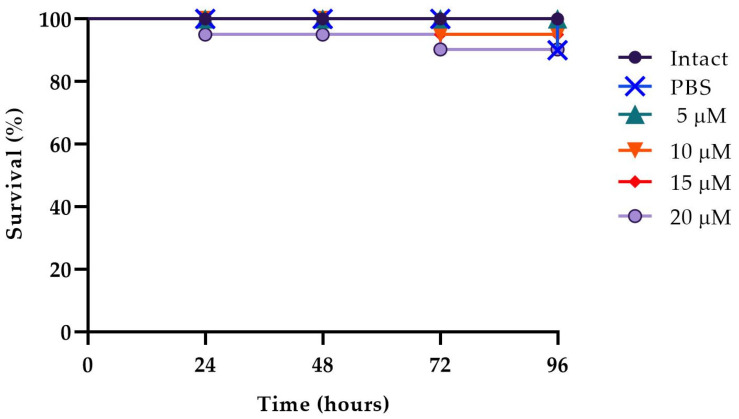
Peptide toxicity on *G. mellonella* larvae treated with WMR-4 at the concentrations of 5 μM, 10 μM, 15 μM, and 20 μM.

**Figure 8 ijms-24-03092-f008:**
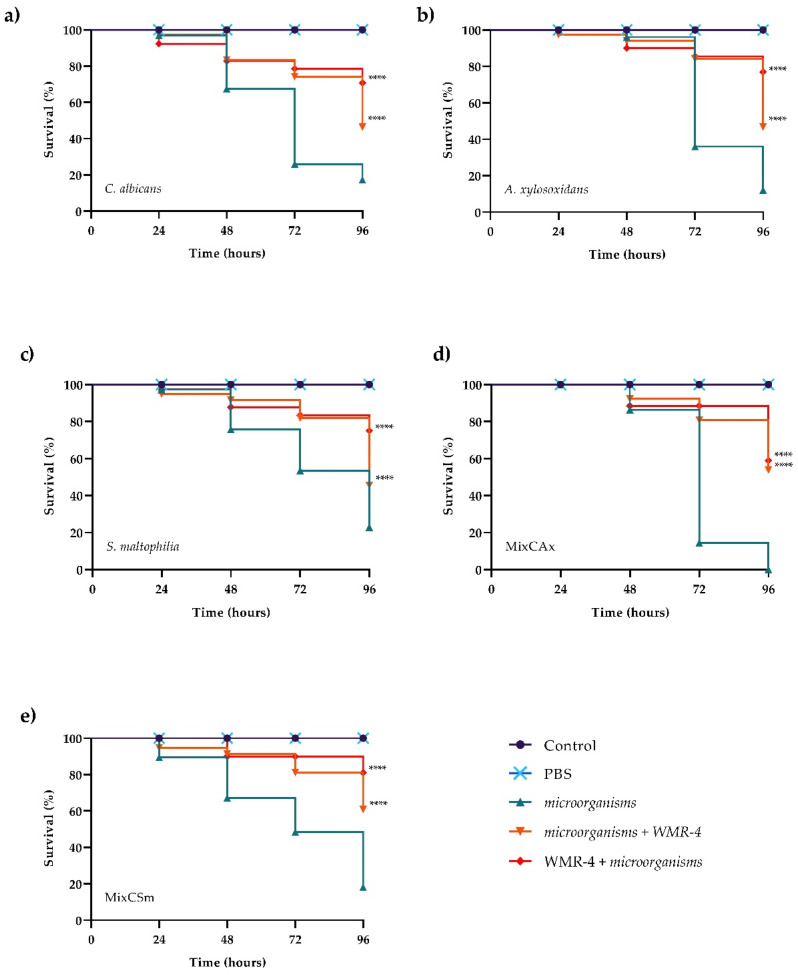
Kaplan–Meier plots of survival curves of *G. mellonella* larvae infected with *C. albicans* (**a**), *A. xylosoxidans* (**b**), *S. maltophilia* (**c**), *CAx* (**d**), *CSm* (**e**). The concentration of microorganisms was 1 × 10^6^ CFU/larva. All groups were treated with 10 μM WMR-4 before or after infection/co-infection. All groups were compared with control (infected or co-infected larvae). In all panels, survival curves of intact larvae and larvae treated with PBS are reported. **** Represents *p*-value < 0.001 (Tukey’s).

**Figure 9 ijms-24-03092-f009:**
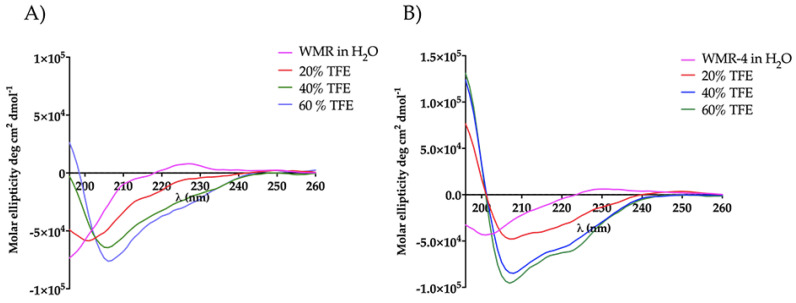
CD spectra of peptides WMR (**A**) and WMR-4 (**B**) in water and in presence of 20%, 40%, and 60% of TFE.

**Figure 10 ijms-24-03092-f010:**
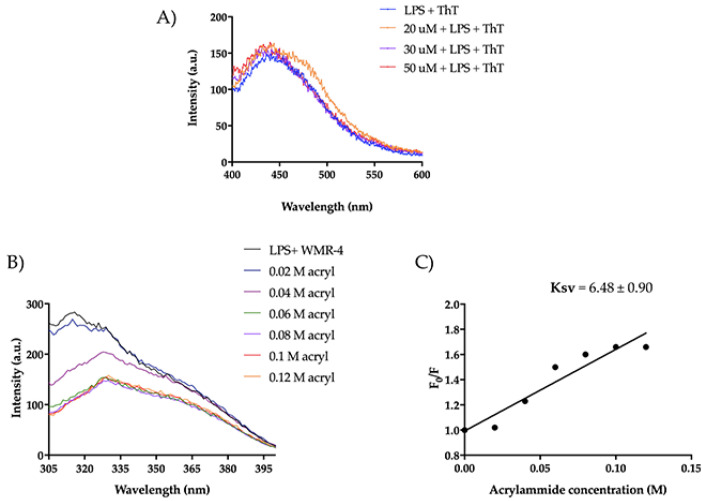
(**A**) ThT emission spectra after the incubation of WMR-4 (20, 30, and 50 μM) with LPS. (**B**) Tryptophan fluorescence spectra for the peptide WMR-4 in LPS during the quenching with acrylamide at different concentrations. (**C**) Stern–Volmer (Ksv) quenching constant of WMR-4 in the presence of LPS.

**Figure 11 ijms-24-03092-f011:**
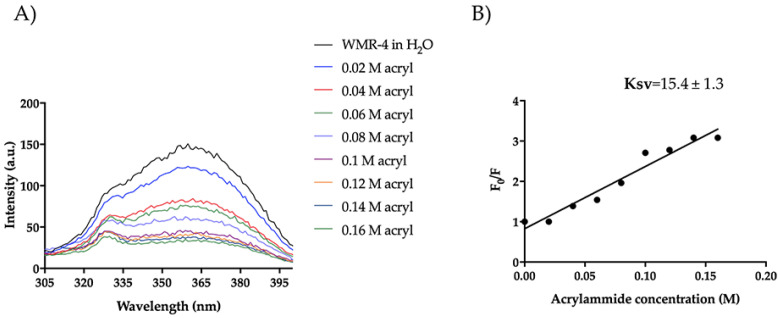
Tryptophan fluorescence spectra for the peptide WMR-4 in water during the quenching with acrylamide at different concentrations (**A**) and Stern–Volmer (Ksv) quenching constant (**B**).

**Figure 12 ijms-24-03092-f012:**
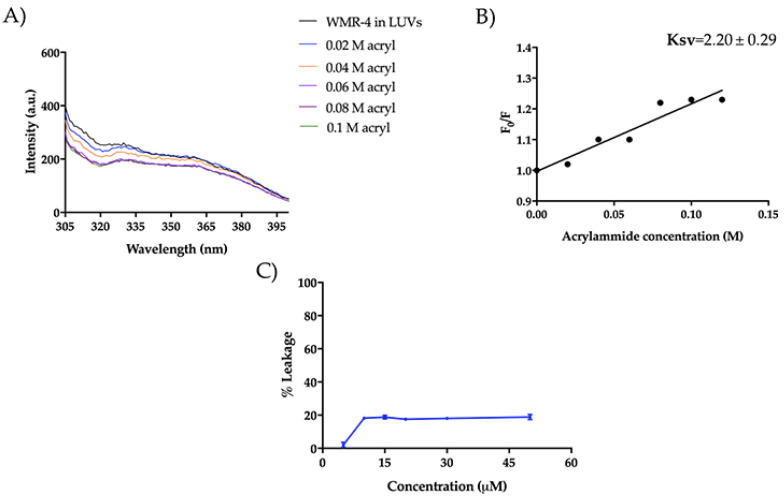
Tryptophan fluorescence spectra for the peptide WMR-4 in LUVs mimicking Gram-negative membranes during the quenching with acrylamide at different concentrations (**A**) and Stern–Volmer (Ksv) quenching constant (**B**). The leakage percentage of LUVs (DOPE/DOPG/CL) induced by WMR-4 (**C**).

**Figure 13 ijms-24-03092-f013:**
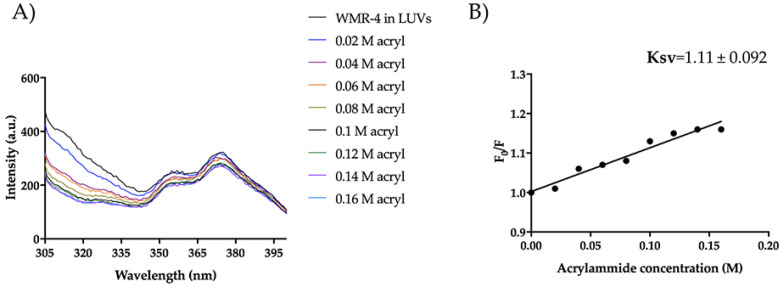
Trp fluorescence spectra for the peptide WMR-4 in LUVs of PE/PC/PI/Erg (**A**) and K_sv_ quenching constant (**B**).

**Table 1 ijms-24-03092-t001:** Identification of serum proteolytic degradation fragments of the peptide WMR.

MW (Da)	Fragment *
1760.15	NH_2_-WGIRRILKYGKRSK-CONH_2_
1443.86	NH_2_-WGIRRILKYGKRS(K)-CONH_2_
1543.86	NH_2_-WGIRRILKYGKR(SK)-CONH_2_
1361.80	NH_2_-(W)GIRRILKYGKRSK-CONH_2_

* Fragments were produced following the proteolytic cutting of the residues (in red) indicated in the brackets.

**Table 2 ijms-24-03092-t002:** Peptide sequences of WMR and its analogues designed in this study. The amino acids used in our study are represented in red.

Name	Sequence
WMR	NH_2_-WGIRRILKYGKRSK-CONH_2_
WMR-1	NH_2_-WGIRRILKYGKRSk-CONH_2_
WMR-2	NH_2_-WGIRRILKYGKRsK-CONH_2_
WMR-3	NH_2_-WAibIRRILKYGKRSKCONH_2_
WMR-4	NH_2_-WAibIRRILKYGKRskCONH_2_
WMR-5	NH_2_-WGIRRILKYGKRSDapCONH_2_

**Table 3 ijms-24-03092-t003:** Minimal inhibitory concentration (MIC) of WMR analogues, FLC, and MEM against species used in our study.

MIC (mM)								
	WMR	WMR-1	WMR-2	WMR-3	WMR-4	WMR-5	FLC (µg/mL)	MEM (µg/mL)
*C. albicans* ATCC 90028	>50	>50	>50	>50	50	>50	16 (48)	-
*A. xylosoxidans* DSM 2402	50	>50	>50	>50	25	25	-	65 (250)
*S. maltophilia* DSM 50170	50	>50	>50	>50	25	25	-	130 (499)

**Table 4 ijms-24-03092-t004:** Primer sequences used in this study.

Micro-Organism	Primer	Primer Sequence (5′->3′)
*A. xylosoxidans*	*Usp-A*	F: CCACGAAGATCCGTACCAGG R: AGGCCTTCTTGCGGTACAC
	*AxyA*	F: CTGGAAGACGGGTCGCAATA R: TGAGTTCACGCAGTTCCACT
	*espF*	F: GCGTTCATCTATCCGGCCAT R: TGATCTCCATGATCGGCAGC
	*rRNA16S*	F: GCAGCAGTGGGGAATTTTGG R: ATTTCACTGCTACACGCGGA
*S. maltophilia*	*smf-1*	F: GGAAGGTATGTCCGAGTCCG R: GCGGGTACGGCTACGATCAGTT
	*Ax21*	F: GGCTACAACGTCGAAATCGC R: ATTCTTCAGCTCGCCGTTCA
	*xanB*	F: TATGCGATCGATGCGTCCAA R: CGAATGCGATCTCTTCGGGA
	*rpfF*	F: CTGGTCGACATCGTGGTG R: TGATCCGCATCATTTCATGC
	*rRNA 16s*	F: ACTGAGACACGGTCCAGACT R: CTTCTTCACCCACGCGGTAT
*C. albicans*	*ERG11*	F: ATTGTTGAAACTGTCATTG R: CCCCTAATAATATACTGATCTG
	*HOG1*	F: GACTTGTGGTCTGTGGGTTG R: ACATCAGCAGGAGGTGAGC
	*ALS3*	F: CTAATGCTGCTACGTATAATT R: CCTGAAATTGACATGTAGCA
	*ACT*	F: AGCCCAATCCAAAAGAGGTATT R: GCTTCGGTCAACAAAACTGG

## Data Availability

Not applicable.

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
