# Peer review of "Myxinidin-Derived Peptide against Biofilms Caused by Cystic Fibrosis Emerging Pathogens"

_ijms, 2023, doi:10.3390/ijms24043092_

Round 1

Reviewer 1 Report

The authors of the paper "Myxinidin-derived peptide against biofilms caused by cystic fibrosis emerging pathogens" describes the possibility of using a engineered peptide WMR-4 against biofilms of bacteria which are otherwise difficult to treat with antibiotics. 

The authors have done a significant amount of work in characterising the effect of their peptide in regards to chemistry and mechanism of action. However, some improvements can be made to the manuscript to make the message clearer and more consise. 

Major comments:

The methods (and choice of method in some cases) need to be improved. 

Section 3.3 (MIC): What is the rationale of using the CLSI guidelines for measuring MIC rather than the EUCAST? In addition, the concentrations which have been chosen are quite strange as the highest concentration tested (50µM) is above cytotoxicity. In addition, why is the concentration of fluconazole written in µM? This is non-standard as MICs for fluconazole and meropenem are presented as µg/mL. 

Section 3.5 (Biofilm): the method for determining the viable biofilm mass might need to be improved. In particular, I am concerned about how the biofilm suspension was prepared. After the biofilms were washed, they should have been sonicated in a water to release the individual bacteria from the biomass. Scraping the plate alone will release clumps of bacteria rather than individual colonies. 

Section 3.8 (moth larvae): I fail to see the logical aspect of using insect larvae to study the possible cytotoxic effect of a drug compound considering its effect in a human host. In addition, why infect larvae with these pathogens and then treat them when the authors are interested in assessing the possible therapeutic effect in the context of cystic fibrosis? It would have made much more sense to assess the cytotoxicity (at this stage) in vitro using primary lung epithelial cells, or immortalised cancerous lung epithelial cells. For example, the cell line A549 could have been easily used to assess cytotoxicity and be used for the infection assays. 

It also appears that the manuscript is missing the discussion section? According to the submission guidelines it is written that for Research Articles: "Introduction, Results, Discussion, Materials and Methods, Conclusions (optional)."

Specific Comments: 

Line 44: I would remove the word "bacterial" in this context as the authors are specifically mentioning that both bacteria and fungi (C. albicans) can grow together 

Lines 54-56: The phrasing of the prevelance of these two bacteria in CF patients can be misinterpreted (is the prevelance of the bacteria within the host or how many CF patients test positive for these bacteria?)

Line 93: the sequence of WMR is unclear, is the first NH2 and last CONH2 an amine and amide, respectively? This is a bit confusing given that inbetween them are the single letter amino acid codes. 

Line 94 (and other locations): please change the phrasing "by some of us" as it is informal. Perhaps use the alternative "previously developed by XXX et al." or "as previous described". 

Line 109: the word "identyfing" is mispelled 

Line 148:  see previous comment regarding "by some of us"

Lines 159-162: The hyphen before CONH2 is missing 

Line 163: It makes logical sense to add the cytotoxicity data here before moving on the MICs. 

Line 168: It does not make sense to have tested up to 50µM of the compound against bacteria if this concentration is known to be cytotoxic. 

Section 2.3: have the authors evaluated the stability of their peptide against specific bacterial or human proteases? '

Section 2.4: See major comment above. In addition, as the results are quite similar, I would merge graphs 3, 4 and 5 into a single graph

Lines 228-244: These graphs have a two tailed error bar, but in figure 2, 4, and 5 it's only showing the upper error bar. Consider changing to match style. 

Line 283: Make the gene ERG1 in italics

Figure 6: This graph is confusingly presented. The data should be shown as compared with the control for each species and target. 

Figure 7: I would consider changing this graph as it is not possible to distinguish between the different symbols 

Figure 8: Similar to the comment above. It is not possible to see the control (overlapping with the PBS group?)

Section 2.7: I would have moved this section earlier before the MICs and moth larvae experiments. It makes more logical sense to explain the structure of the peptide when the amino acid sequence was given.

Section 2.8: Similar to section 2.7, keep the chemistry sections together before moving onto the biological. 

Line 359: Why did the authors go for concentrations above cytotoxicity? 

Section 2.9: Did the authors assess the possible lysis of a human cell membrane model such as a POPC:cholesterol model?

Figure 12: The figure legend heading for subfigure C is missing

Table 4: The primers used for genes from C. albicans are missing.

Author Response

Major comments:

The methods (and choice of method in some cases) need to be improved. 

Section 3.3 (MIC): a) What is the rationale of using the CLSI guidelines for measuring MIC rather than the EUCAST? b) In addition, the concentrations which have been chosen are quite strange as the highest concentration tested (50µM) is above cytotoxicity. c) In addition, why is the concentration of fluconazole written in µM? This is non-standard as MICs for fluconazole and meropenem are presented as µg/mL. 

Reply: We thank the reviewer for his/her positive comments and useful suggestions to improve the manuscript. a) Both methods are based on broth dilution and methodological differences are minimal considering the fact that these are preliminary experiments performed on collection strains and it is not a clinical study. b) As reported in the manuscript, concentrations tested on G. mellonella are sub-MIC concentrations obtained from biofilm studies. c) We choose to write the concentration of fluconazole in µM with the scope to uniform all measurement units but we add also the concentration in µg/mL.

Section 3.5 (Biofilm): the method for determining the viable biofilm mass might need to be improved. In particular, I am concerned about how the biofilm suspension was prepared. After the biofilms were washed, they should have been sonicated in a water to release the individual bacteria from the biomass. Scraping the plate alone will release clumps of bacteria rather than individual colonies. 

Reply: We thank the reviewer for his/her comment. We agree with the reviewer and in fact, during the biofilm formation, we vigorously vortexed the scraped biofilms after the biofilms were washed. In this regard, we added this sentence to the text: “ xxxxxx” (see line 526).

Section 3.8 (moth larvae): I fail to see the logical aspect of using insect larvae to study the possible cytotoxic effect of a drug compound considering its effect in a human host. In addition, why infect larvae with these pathogens and then treat them when the authors are interested in assessing the possible therapeutic effect in the context of cystic fibrosis? It would have made much more sense to assess the cytotoxicity (at this stage) in vitro using primary lung epithelial cells, or immortalised cancerous lung epithelial cells. For example, the cell line A549 could have been easily used to assess cytotoxicity and be used for the infection assays. 

Reply: We thank the reviewer for his/her comment. We chose this model in our experimental study because G. mellonella larvae are an alternative in vivo model that has gained increasingly space due to its low cost, low biological risk and the similarity of its immune system to mammals’ innate immune response, which makes their use even more desirable in predicting toxicity in humans (see references: doi: 10.1080/21505594.2019.1621649; doi: 10.1080/21505594.2015.1135289. Furthermore, we are working on assessing the usefulness of our peptides in cystic fibrosis and the study of their effect on lung cells infected with bacteria will be the objective of future studies.

It also appears that the manuscript is missing the discussion section? According to the submission guidelines it is written that for Research Articles: "Introduction, Results, Discussion, Materials and Methods, Conclusions (optional)."

Reply: We thank the reviewer for his/her comment. According to the journal guidelines, it is possible to report “result and discussion”  in the same section.

Specific Comments: 

Line 44: I would remove the word "bacterial" in this context as the authors are specifically mentioning that both bacteria and fungi (C. albicans) can grow together 

Reply: We thank the reviewer for his/her comment. We modified the text accordingly.

Lines 54-56: The phrasing of the prevelance of these two bacteria in CF patients can be misinterpreted (is the prevelance of the bacteria within the host or how many CF patients test positive for these bacteria?)

Reply: We thank the reviewer for his/her comment. We modified the text accordingly.

Line 93: the sequence of WMR is unclear, is the first NH2 and last CONH2 an amine and amide, respectively? This is a bit confusing given that in between them are the single letter amino acid codes. 

Reply: We thank the reviewer for his/her comment. In the WMR sequence, -NH2 and -CONH2 represent the amino and amide groups, respectively. This is the way usually peptides are reported.  

Line 94 (and other locations): please change the phrasing "by some of us" as it is informal. Perhaps use the alternative "previously developed by XXX et al." or "as previous described". 

Reply: We thank the reviewer for his/her comment. We modified the text accordingly.

Line 109: the word "identyfing" is mispelled 

Reply: We thank the reviewer for his/her comment. We modified the text accordingly

Line 148:  see previous comment regarding "by some of us".

Reply: We thank the reviewer for his/her comment. We modified the text accordingly

Lines 159-162: The hyphen before CONH2 is missing 

Reply: We thank the reviewer for his/her comment. We added them in the manuscript.

Line 163: It makes logical sense to add the cytotoxicity data here before moving on the MICs. 

Reply: We thank the reviewer for his/her comment. The first goal of our study was to test antibiofilm activity of the peptide WMR-4 at sub-MIC concentrations and then we tested its toxicity in vivo at the concentrations determined.

Line 168: It does not make sense to have tested up to 50µM of the compound against bacteria if this concentration is known to be cytotoxic. 

Reply: We thank the reviewer for his/her comment. We tested the cytotoxicity of WMR-4 up to 20 µM because it showed a potent antibiofilm activity in this concentration range.

Section 2.3: have the authors evaluated the stability of their peptide against specific bacterial or human proteases?.

Reply: We thank the reviewer for his/her comment. In this work, we tested the stability of our peptide in presence of fetal bovine serum. The peptide stability in presence of human proteases will be surely evaluated in future studies.

Section 2.4: See major comment above. In addition, as the results are quite similar, I would merge graphs 3, 4 and 5 into a single graph

Reply: We thank the reviewer for his/her comment. We believe that in this way it could be easier to understand, also because figures reported different data.

Lines 228-244: These graphs have a two tailed error bar, but in figure 2, 4, and 5 it's only showing the upper error bar. Consider changing to match style. 

Reply: We thank the reviewer for his/her comment.  We matched the style for figures 2,4 and 5.

Line 283: Make the gene ERG1 in italics

Reply: We thank the reviewer for his/her comment. We modified it in the manuscript.

Figure 6: This graph is confusingly presented. The data should be shown as compared with the control for each species and target. 

Reply: We thank the reviewer for his/her comment. The REST software used for analysing the expression results directly reports fold variations respect to untreated controls (in our case samples of biofilm growth without WMR-4).

Figure 7: I would consider changing this graph as it is not possible to distinguish between the different symbols 

Reply: We thank the reviewer for his/her comment. We improved the style of figure 7.

Figure 8: Similar to the comment above. It is not possible to see the control (overlapping with the PBS group?)

Reply: We thank the reviewer for his/her comment. We modified figure 8.

Section 2.7: I would have moved this section earlier before the MICs and moth larvae experiments. It makes more logical sense to explain the structure of the peptide when the amino acid sequence was given.

Reply: We thank the reviewer for his/her comment. Our aim was to improve the peptide stability by designing five WMR-derived peptides by replacing the amino acids recognized by proteases. The first step was to evaluate the effect of the replacement on the antimicrobial activity and cytotoxicity. After this first evaluation, we explored the impact of this modification on peptide structure comparing the best compound with its native peptide.  

Section 2.8: Similar to section 2.7, keep the chemistry sections together before moving onto the biological. 

Reply: We thank the reviewer for his/her suggestion. We commented our choice above (section 2.7.).

Line 359: Why did the authors go for concentrations above cytotoxicity? 

Reply: We thank the reviewer for his/her comment. We used concentration above cytotoxicity in biophysical studies in order to better investigate the mode of action of peptide WMR-4 using liposomes mimicking bacterial and fungal membranes.

Section 2.9: Did the authors assess the possible lysis of a human cell membrane model such as a POPC:cholesterol model?

Reply: We thank the reviewer for his/her comment. In this work, we focused on the mode of action of the peptide WMR-4 using models mimicking bacterial and fungal membranes, but we will explore its mode of action in presence of POPC:cholesterol model in the next studies. By the way, we previously assessed the activity of an analogue (WMR) on liposomes mimicking eukaryotic cells and we found that no activity was performed on those cells, further supporting the lack of toxicity.  

Figure 12: The figure legend heading for subfigure C is missing

Reply: We thank the reviewer for his/her comment. We added it.

Table 4: The primers used for genes from C. albicans are missing.

Reply: We thank the reviewer for his/her comment. We added the primers in the table 4.

Reviewer 2 Report

Authors have made wonderful efforts and have been successful to make a contribution to the research area of antimicrobial peptides. The methodology is chosen appropriately for laboratory work. The abstract and introduction parts are also well-written to provide a basic idea of conducted research to the readers. But here some minor improvements are needed.

1.     I don't find any notes on Table 1 and Table 2 marked in red. I think it would be easier for the reader to understand by providing an explanation.

2.     The survival rates between several groups are more difficult to distinguish and it is proposed to modify the graph in Figure 7.

3.     In section 2.7, it is proposed to add a table of secondary structures to show more accurately the proportions of the various structures.

Author Response

Authors have made wonderful efforts and have been successful to make a contribution to the research area of antimicrobial peptides. The methodology is chosen appropriately for laboratory work. The abstract and introduction parts are also well-written to provide a basic idea of conducted research to the readers. But here some minor improvements are needed.

  1. I don't find any notes on Table 1 and Table 2 marked in red. I think it would be easier for the reader to understand by providing an explanation.

Reply: We thank the reviewer for his/her comment. We added the note in the tables 1 and 2.

  1. The survival rates between several groups are more difficult to distinguish and it is proposed to modify the graph in Figure 7.

Reply: We thank the reviewer for his/her comment. We modified figure 7 in the manuscript.

  1. In section 2.7, it is proposed to add a table of secondary structures to show more accurately the proportions of the various structures.

Reply: We thank the reviewer for his/her comment, we add the percentage of a-helical conformation to the maximum concentration of TFE.

Reviewer 3 Report

Bellavita et al. present a very interesting and diverse study on a myxinidin-derived AMP. They design new more stable AMPs, test them in vitro and in vivo, and, in addition, perform studies to explain the mecanism of action using liposomes. Here are several points to improve:

- English language editing is required

- The beginning of the Introduction part is poorly structured and repetitive

- The biofilms need to be introduced to the reader

- Line 94, « some of us » - please clarify

- Figure 2: WMR-4 stability decreases to 30% after 16h at 37°C. Will it be enough to inhibit/eradicate the biofilms? Please add some discussion.

- It would be nice to have some microscopy images of the biofilms, in addition to the quantitative results

- Figure 6: is there a particular reason why 10 µM and not 20 µM were chosen for RT-qPCR experiment? Please explain.

- Tryptophan quenching experiment should be introduced to the readers

Author Response

Bellavita et al. present a very interesting and diverse study on a myxinidin-derived AMP. They design new more stable AMPs, test them in vitro and in vivo, and, in addition, perform studies to explain the mechanism of action using liposomes. Here are several points to improve:

- English language editing is required

Reply: We thank the reviewer for his/her comment. We had the manuscript revised by a native English collegue

- The beginning of the Introduction part is poorly structured and repetitive

Reply: We thank the reviewer for his/her comment and suggestion. We modified the beginning of the introduction.

- The biofilms need to be introduced to the reader

Reply: We thank the reviewer for his/her suggestion. We added it in the introduction section.

- Line 94, « some of us » - please clarify

Reply: We thank the reviewer for his/her comment. We modified it.

- Figure 2: WMR-4 stability decreases to 30% after 16h at 37°C. Will it be enough to inhibit/eradicate the biofilms? Please add some discussion.

Reply: We thank the reviewer for his/her comment. WMR-4 stability decreases to 30% after 16h, but the effect is due to intact peptide, that change in the time

- It would be nice to have some microscopy images of the biofilms, in addition to the quantitative results

Reply: We thank the reviewer for his/her comment and this may be the subject of a future study.

- Figure 6: is there a particular reason why 10 µM and not 20 µM were chosen for RT-qPCR experiment? Please explain.

Reply: We thank the reviewer for his/her comment. We chose 10 µM because it is the concentrations that give about 50% inhibition for all microorganisms.

- Tryptophan quenching experiment should be introduced to the readers

Reply: We thank the reviewer for his/her comment. We added it in the section 2.8.

Reviewer 4 Report

This work synthesized a serum-stable version of the peptide WMR (WMR-4), which was able to eradicate both mono and dual-species biofilms of C. albicansS. maltophilia, and A. xylosoxidans. I find this manuscript important for future antibacterial research and the treatment of cystic fibrosis. The topic falls within the scope of the IJMS journal. I recommend acceptance once one minor comment below is addressed.

1.    In section 2.4, further explanation is needed to explain why biofilm growth was selected for 24 and 48 hours. At this stage, dispersion might be dominant, which can make it hard to quantify the amount of biofilm formed by using crystal violet staining.

Author Response

Comments and Suggestions for Authors

This work synthesized a serum-stable version of the peptide WMR (WMR-4), which was able to eradicate both mono and dual-species biofilms of C. albicansS. maltophilia, and A. xylosoxidans. I find this manuscript important for future antibacterial research and the treatment of cystic fibrosis. The topic falls within the scope of the IJMS journal. I recommend acceptance once one minor comment below is addressed.

In section 2.4, further explanation is needed to explain why biofilm growth was selected for 24 and 48 hours. At this stage, dispersion might be dominant, which can make it hard to quantify the amount of biofilm formed by using crystal violet staining.

Reply: We thank the reviewer for his/her comment. Considering the different time of growing of the three microorganism, we would like to show also a characterization of mature biofilms, even if, as noticed by reviewer, it is hard to quantify the mount of biofilm formed by CV, we count CFUs in order to have the vital biomass also at 48 h.

Round 2

Reviewer 1 Report

Bellavita et al. have made improvement to their manuscript entitled "Myxinidin-derived peptide against biofilms caused by cystic fibrosis emerging pathogens". 

While some questions still remain regarding the effect of their peptide against human membrane models, human lung cell toxicity, etc., I am sure that they will answer these in their future research.